# EVERYONE COUNTS: FAIR AND ACCURATE HETEROGENEOUS FEDERATED LEARNING WITH RESOURCE-ADAPTIVE MODEL MODULATION

## ABSTRACT

In the practical implementation of federated learning (FL), a major challenge arises from the presence of diverse and heterogeneous edge devices in real-world scenarios, each equipped with varying computational resources. The conventional FL approaches, operating under the assumption of uniform model capacity, face a dilemma. They can opt for a large global model, but this may not be feasible on resource-constrained devices, resulting in issues of fairness and training biases. Conversely, they can choose a small global model, but this compromises its ability to represent complex patterns due to limited capacity. In this paper, we present a novel approach called Dynamic Federated Learning (DynamicFL). It employs structural re-parameterization to achieve adaptable local model modulation and seamless knowledge transfer across a diverse set of heterogeneous models. DynamicFL ensures equitable treatment of all clients, empowering them to actively participate in the learning process with their full computational potential, thereby fostering sustainability within the FL ecosystem. Extensive experimental results validate that DynamicFL surpasses state-of-the-art techniques, including knowledge distillation and network pruning-based methods, in achieving significantly higher test accuracy in the context of heterogeneous FL.

## 1 INTRODUCTION

The success of deep learning heavily relies on large-scale training datasets (LeCun et al., 2015). However, in many privacy-sensitive scenarios, such as finance (Yang et al., 2019; Long et al., 2020) and biomedicine (Xu et al., 2021), amassing a sufficiently large centralized training dataset is often impractical due to stringent privacy regulations such as GDPR (GDPR, 2016) and ADPPA (ADPPA, 2022). Federated Learning (FL), as an emerging privacy-aware distributed learning paradigm, addresses this challenge by allowing multiple parties to collaboratively train a model without sharing their raw data. This key attribute has garnered increasing attention from both academia and industry.

One of the primary challenges for the practical deployment of FL is the vast heterogeneity of edge devices in real-world settings (Zhou et al., 2019). These devices range from wearable gadgets and mobile phones to edge servers, each with varying computing power, memory capacities, and communication bandwidth. Despite this diversity, most FL paradigms (McMahan et al., 2017) assume a uniform model capacity, where local models share the same architecture as the global model. However, this approach hinders the participation of clients with limited computational resources, as they may be unable to afford the cost of local training. Furthermore, resource budgets are often closely tied to the demographic and socioeconomic characteristics of owners. The exclusion of such clients gives rise to fairness issues in terms of participation and introduces training bias due to the absence of unique data from these clients. To address the heterogeneity of edge devices, it is desirable to develop an FL framework with heterogeneous local models that can adapt to varying neural network complexities based on their computational resource budgets. However, this introduces a new challenge: *how can knowledge be effectively exchanged across these heterogeneous local models to derive a single global inference model?*

To tackle this problem, several existing works have focused on either knowledge distillation-based or network pruning-based approaches. Knowledge distillation, a popular technique for transferring

knowledge across different network structures (Hinton et al., 2015), has been incorporated into the field of Federated Learning (FL) to address the challenge of model heterogeneity. For instance, FedGKT (He et al., 2020) was introduced to train small convolutional neural networks (CNN) on edge nodes and periodically transfer their knowledge via distillation to a larger CNN on the server side. To enhance knowledge transfer, FedDF (Lin et al., 2020) utilizes a public dataset (e.g., unlabeled data or artificially generated examples) to aggregate knowledge from diverse client models. DSFL (Itahara et al., 2021) proposes a distillation-based semi-supervised FL algorithm that exchanges the outputs of local models among mobile devices to label samples of a public dataset, which are then used for further training of the local models. FedET (Cho et al., 2022) employs public unlabeled data to facilitate a bi-directional ensemble knowledge transfer between the server and client models.

Pruning-based approaches attempt to adjust the size of CNNs by modifying their width or depth, known to be effective in controlling model capacity. HeteroFL (Diao et al., 2021) and FjORD (Horváth et al., 2021) opt to vary the width of hidden channels to dynamically adapt model size. These methods yield local models with significantly fewer parameters while still belonging to the same model class as the global model. As a result, aggregation is achieved by averaging parameters of the overlapping sub-networks across different models. However, as observed by (Mei et al., 2022), the performance improvement of HeteroFL and FjORD is inconsistent, suggesting that sharing parts of network widths across different clients may not effectively transfer knowledge. In (Mei et al., 2022), the adjustment of width for resource adaptation is considered. Instead of pruning the global model into local ones, it formulates networks at different capacities as linear combinations of one unified set of parameters. InclusiveFL (Liu et al., 2022) chooses to reduce the number of parameters in local models by decreasing the depth of the network. It accomplishes knowledge transfer by sharing the shallow bottom layers of the largest model with other smaller models, and employs a momentum distillation technique to enhance the effectiveness of knowledge transfer. However, compared to reducing width, a reduction in depth is less effective in reducing parameters and memory footprint during inference (Mei et al., 2022).

Although serving as popular strategies for handling heterogeneous FL, it is well-known that knowledge distillation cannot losslessly transfer knowledge cross diverse network structures (Huang et al., 2022a), leading to limited performance; in pruning-based strategy, sub-network and complete network may have different behaviors, leading to mismatch of feature spaces. Except knowledge distillation based and network pruning based approaches, is there any other strategy that can develop opportunities for all the clients to achieve their full potential? To answer this question, two issues should be carefully considered: 1) *how to design resource-adaptive network architecture for each client so as to exert all their strength?* 2) *how to project heterogeneous local models into a uniform model architecture so as to facilitate the aggregation operation?*

In this paper, we offer a new line for heterogeneous FL and design a unified framework to address the above concerns. Specifically, we propose dynamic federated learning (DynamicFL) framework, which conducts structural re-parameterization to achieve resource-adaptive local models modulation and lossless knowledge transfer across heterogeneous local models, as illustrated in Fig. 1. The key steps of DynamicFL include heterogeneous local training and homogeneous global aggregation:

– **Heterogeneous Local Training:** For each client, according to the local computational resource, DynamicFL adaptively expands local models to multi-branches with appropriate capacity from the VGG-style plain global model by re-parameterization. Every client works upon the modulated local model in full operational capability, thus can achieve high accuracy of local training. Moreover, different from the vanilla structural re-parameterization (Ding et al., 2021a;b) in which all candidate operations are re-parameterized, we dynamically adjust computation workloads by choosing the operations with the significant contributions to the performance, so as to achieve the trade-off between training accuracy and efficiency.

– **Homogeneous Global Aggregation:** After local training, the heterogeneous local models are transformed back to the original global model structure by re-parameterization. In this way, all the uploaded local model have the same structure, over which the aggregation operation can be done without any knowledge transfer consumption. Since no complicated knowledge distillation is performed on the server side, the aggregation operation can be done rapidly.

The main contributions of this paper are summarized as follows:

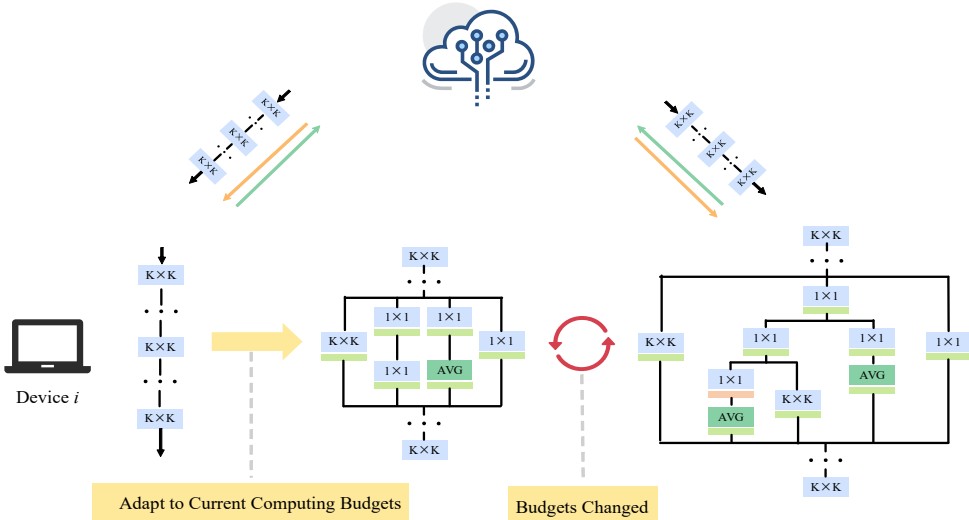

Figure 1: Overview of DynamicFL.

**– Fair Clients Participation:** Our method fairly treats all clients, no matter strong or indigent, and encourages them actively participating in the learning process with their full operational capability. Moreover, the knowledge of the global model can be fully transferred to even indigent clients, without the consumption as knowledge distillation based and network pruning based approaches. In our scheme, every client counts and gives their best. This property is conducive to the sustainability of the FL ecosystem.

**– Improved Model Accuracy:** Attributing to that all clients works at full capacity and knowledge is losslessly transferred across heterogeneous models, our method achieves improved model accuracy. As demonstrated by experimental results, DynamicFL achieves much better performance than the state-of-the-art knowledge distillation based and network pruning based heterogeneous FL methods.

**– Convergence Guarantee:** We theoretically provide a convergence guarantee for our method and conduct extensive experiments to verify the effectiveness of the proposed method in learning an accurate global model from heterogeneous clients in the FL framework. Please refer to Appendix for more details.

## 2 METHODOLOGY

### 2.1 STRUCTURAL RE-PARAMETERIZATION

The core idea of structural reparameterization is that various architectures can be transformed into each other through equivalent parameter transformations. During training, it employs complex multi-branch structures, which are equivalently merged into a single-branch structure during inference. This property has led to its widespread use in lightweight model deployment. (Ding et al., 2021b) enhances VGG-style networks by expanding the $3 \times 3$ Convolutional layer into three branches, namely $3 \times 3$ Conv, $1 \times 1$ Conv, and residual connection, during training and re-parameterizing it back to the original $3 \times 3$ Conv during inference. (Ding et al., 2021a) enriches CNNs by introducing six equivalent transformations of re-parameterization, thereby diversifying the types of expanding branches, and unifying them into a universal building block.

As indicated by (Ding et al., 2019; Huang et al., 2022b), the 2D convolutions hold the property of additivity:

$$\boldsymbol{I} \circledast \boldsymbol{F}^{(1)} + \boldsymbol{I} \circledast \boldsymbol{F}^{(2)} = \boldsymbol{I} \circledast \left( \boldsymbol{F}^{(1)} + \boldsymbol{F}^{(2)} \right). \tag{1}$$

where $\boldsymbol{I}$, $\boldsymbol{F}^{(1)}$ and $\boldsymbol{F}^{(2)}$ are the input and kernels, respectively. The above equation is satisfied even with different kernel sizes. Some widely used operations in CNN—average pooling and batch normalization—can be converted into convolution operation.

The above additivity property ensures that a single convolution can be equivalently transformed to multi-branch operations, and vice versa. This is particularly attractive for heterogeneous FL: we can equivalently transform the plain global model to multi-branch local models, which are good at training and deliver high accuracy; and equivalently transform multi-branch local models to the plain global model, which facilitate parameters averaging in aggregation. The equivalent transformations of operations guarantee lossless knowledge transfer, since the model outputs are not changed along with the network structure adjustment.

## 2.2 DYNAMIC FEDERATED LEARNING

We consider federated learning across $N$ heterogeneous edge clients with diverse computational capabilities. Each client $i$ can only access to its own private dataset $\mathcal{D}_i = \{(\mathbf{x}_j^i, y_j^i)\}$ where $\mathbf{x}$ and $y$ denote the input features and corresponding class labels, respectively. In the following, we elaborate the main steps ofresource-adaptive dynamic federated learning scheme (DynamicFL). The workflow of DynamicFL is offered in Algorithm 1.

### 2.2.1 INITIALIZATION

In the first round, once the global model is received, the client $i$ expands the $K \times K$ convolution layers to diverse branches adopted in DBB (Ding et al., 2021a) from bottom to up according to their computational resource:

$$\zeta_i^{(0)} \leftarrow \text{REP}(\omega_g^{(0)}). \tag{2}$$

where $\text{REP}(\cdot)$ represents the re-parameterization operation, and the $\zeta_i^{(0)}$ has the same the output as the global model $\omega_g^{(0)}$ but has a bigger model size according to $r_i$. The stronger the client, the more layers it expands. Upon the derived local model $\zeta_i^{(0)}$, we perform local training for $E$ epochs:

$$\begin{aligned}
\zeta_i^{(0)} &\leftarrow \zeta_i^{(0)} - \eta \nabla \ell(\zeta_i^{(0)}; \mathcal{D}_i), \\
S_i^l &\leftarrow \nabla \ell(\zeta_i^{(0)}; \mathcal{D}_i).
\end{aligned} \tag{3}$$

We record the gradient of the last epoch as $S_i^l$, which reflects the sensitivity of each branch of the local model to local knowledge. The local model is then transformed back to the original plain structure and uploaded to the server. Since all uploaded local models have the same structure as the global model, we can perform the aggregation operation to derive the updated global model $\omega_g^{(0)}$.

### 2.2.2 HETEROGENEOUS LOCAL TRAINING

The new global model is distributed to local clients, upon which the next round of local training is conducted. Considering local clients have diverse and limited computational resources, we dynamically adjust computation workloads by choosing the operations with the significant contributions to the performance instead of re-parameterizing all the candidate operations as done in DBB. The resource-adaptive models modulation is tailored according the local and global gradient information of FL.

Specifically, in the $t$-th round, relying on the new global model $\omega_g^{(t-1)}$ and the stale local model $\zeta_i^{(t-1)}$ in the last round, we conduct structural re-parameterization to derive a temporary local model $\epsilon_i^{(t)}$ that has the same output as $\omega_g^{(t-1)}$ while with the same structure as $\zeta_i^{(t-1)}$:

$$\epsilon_i^{(t)} \leftarrow \text{REP}(\omega_g^{(t-1)}, \zeta_i^{(t-1)}). \tag{4}$$

We perform local training on $\epsilon_i^{(t)}$ along with the private dataset $\mathcal{D}_i$ for only one epoch, and obtain the gradient information $S_i^g$:

$$\begin{aligned}
\epsilon_i^{(t)} &\leftarrow \epsilon_i^{(t)} - \eta \nabla \ell(\epsilon_i^{(t)}; \mathcal{D}_i), \\
S_i^g &\leftarrow \nabla \ell(\epsilon_i^{(t)}; \mathcal{D}_i),
\end{aligned} \tag{5}$$

where $S_i^g$ reflects the sensitivity of each branch of the local model to the global knowledge, *i.e.*, the aggregation result of all participants. $S_i^g$ and $S_i^l$ offer useful cues to reflect the contribution of branches to the global aggregation. According to them, we dynamically evolve the network structures

of the local model to remove redundant operations with little contribution while further expanding important operations with significant contribution, and obtain the new local model $\zeta_i^{(t)}$:

$$\zeta_i^{(t)} \leftarrow \text{DYMM}(\epsilon_i^{(t)}; S_i^g, S_i^l). \tag{6}$$

The details of $\text{DYMM}(\cdot)$ for how to use $S_i^g, S_i^l$ to perform dynamic model modulation can be found in Section 2.3. Client $i$ then performs local training along with its private data on the new local model $\zeta_i^{(t)}$ with full operational capability:

$$\zeta_i^{(t)} \leftarrow \zeta_i^{(t)} - \eta \nabla \ell(\zeta_i^{(t)}; \mathcal{D}_i). \tag{7}$$

where the number of epochs executed is $E - 1$, since one epoch has already been done in Eq. (5). The gradient information of the last epoch is recorded as the updated $S_i^l$:

$$S_i^l \leftarrow \nabla \ell(\zeta_i^{(t)}; \mathcal{D}_i). \tag{8}$$

It is worth mentioning a possible scenario: in this round, client $i$ may have less computational budget than the last round due to that some other routines are executing, leading to Client $i$ cannot afford the local model of the last round. In this case, the steps corresponding to Eq. (4), (5) and (6) cannot be performed. Instead, we conduct re-parameterization based on $S_i^l$ only to derive the new model:

$$\zeta_i^{(t)} \leftarrow \text{REP}(\omega_g^{(t-1)}, S_i^l). \tag{9}$$

Then local training on $\zeta_i^{(t)}$ with $E$ epochs is performed as Eq. (7).

### 2.2.3 Homogeneous Global Aggregation

Once Client $i$ completes the local training, the local model $\zeta_i^{(t)}$ is transformed back to the original global model structure by re-parameterization:

$$\omega_i^{(t)} \leftarrow \text{REP}(\zeta_i^{(t)}, \omega_g^{(t-1)}), \tag{10}$$

$\omega_i^{(t)}$ has the same output as the local model $\zeta_i^{(t)}$ while with the same structure as the global model $\omega_g^{(t-1)}$. After the central server receives the uploaded $\{\omega_i^{(t)}\}_{i=1}^N$ from all clients, the global aggregation can be done since $\{\omega_i^{(t)}\}_{i=1}^N$ share the same network structure:

$$\omega_g^{(t)} \leftarrow \frac{1}{N} \sum \omega_i^{(t)}. \tag{11}$$

The equivalent transformations of operations in re-parameterization guarantee lossless knowledge transfer across heterogeneous local models. The central server then distributes the new global model $\omega_g^{(t)}$ to local clients to start the next round of local training.

### 2.3 Dynamic Model Modulation

In this subsection, we present the detail of $\text{DYMM}(\cdot)$ operator to perform dynamic model modulation based on the local knowledge from the local private dataset and the global knowledge from the global aggregation.

We dynamically identify the important operations and redundant operations of local models according to the gradient information. Some literature (Lee et al., 2018; Wang et al., 2020) has shown that the gradient of each weight of the model in the training process can effectively reflect the sensitivity of the weight to data. Inspired by (Tanaka et al., 2020; Huang et al., 2022b), we use the following metric to measure the saliency of each weight:

$$\mathcal{S}(\theta_i) = \frac{\partial \ell}{\partial \theta}(\theta_i), \tag{12}$$

where $\theta_i \in \boldsymbol{\theta}$ is the parameter of the model.

The local knowledge in FL is easy to obtain during local training. In the $(t-1)$-th round, we record the last local epoch gradient information by Eq. (8) and (12):

$$S_i^l(\boldsymbol{\theta}^k) = \sum_j^n \frac{\partial \ell}{\partial \theta_j^k} \odot \theta_j^k, \tag{13}$$

where $\boldsymbol{\theta}^k$ is the $k$-th branch in the local model $\zeta_i^{(t-1)}$, and $\theta_j^k$ is the $j$-th parameter of the branch $k$. $S_i^l(\boldsymbol{\theta}^k)$ is leveraged for dynamic model modulation in the the next round.

The global knowledge in FL is implicitly encoded into the global model, which is the aggregation result of all participating clients. To extract global information, according to the received global model $\omega_g^{(t-1)}$ in the $t$-th round and the last round local model $\zeta_i^{(t-1)}$, re-parameterization is conducted to get the temporary model $\epsilon_i^t$ through Eq. (4). Here $\epsilon_i^t$ is a re-parameterized model of $\omega_g^{(t-1)}$, which has the same structure as $\zeta_i^{(t-1)}$. We use $\epsilon_i^t$ to perform one-epoch local training with the private dataset, from which the derived gradient information reflects the sensitivity of the branches in the local model $\zeta_i^{(t-1)}$ to the global information:

$$S_i^g(\boldsymbol{\psi}^k) = \sum_j^n \frac{\partial \ell}{\partial \psi_j^k} \odot \psi_j^k, \tag{14}$$

where $\boldsymbol{\psi}^k$ is the $k$-th branch in the local model $\epsilon_i^{(t)}$, and $\psi_j^k$ is the $j$-th parameter of branch $k$.

We regard the branches with both small $S_i^l$ and $S_i^g$ as redundant ones; the branches with a small $S_i^l$ but large $S_i^g$ as important ones since they are sensitive to the global knowledge; the rest branches as common ones. In Eq. (6), we first merge redundant branches into common branches to reduce the size of the local model:

$$\boldsymbol{\psi}^{(\text{common'})} \leftarrow \boldsymbol{\psi}^{(\text{common})} + \boldsymbol{\psi}^{(\text{redundant})}, \tag{15}$$

where $\boldsymbol{\psi}^{(\text{common'})}$ is the new common branch after merging; $\boldsymbol{\psi}^{(\text{common})}$ and $\boldsymbol{\psi}^{(\text{redundant})}$ represent common and redundant branches, respectively. Then we expand the important branches to adapt the device's computing capabilities:

$$\boldsymbol{\psi}^{(\text{important'})} \leftarrow \boldsymbol{\psi}^{(\text{important})} - (\boldsymbol{\psi}^{(1)} + \cdots + \boldsymbol{\psi}^{(n)}), \tag{16}$$

where $\boldsymbol{\psi}^{(\text{important'})}$ are the expanded parameters of the important branch. In order to ensure the same output, we need to subtract the parameters of the new branch $\{\boldsymbol{\psi}^{(i)}\}_{i=1}^n$ from the parameters of the original important branch $\boldsymbol{\psi}^{(\text{important})}$. The parameters of the new branch are randomly generated.

## 3 EXPERIMENTS

**Datasets and Models.** We evaluate the performance of DynamicFL under two popular benchmark datasets: CIFAR-10, CIFAR-100 and a medical image dataset DermaMNIST (Yang et al., 2021). DermaMNIST is a dataset consisting of multi-source dermatoscopic images of common pigmented skin lesions. It contains 10,015 dermatoscopic images categorized into 7 different diseases, with an image size of $3 \times 28 \times 28$. We adopt ResNet-18, GoogLeNet, and LeNet-5 to evaluation our method. We compare our method with knowledge distillation based methods and pruning based methods. For the former, we use the same local models as our method and select a largest local model as the global model on the central server to ensure a fair comparison. For the latter, they only support pruned versions of the global model as local models. Therefore, they could not adopt exactly the same settings as our method. we adopt the same global models as distillation based methods as their global model and the corresponding pruned models as local models.

**Data & Model Heterogeneity.** In line with prior works (Liu et al., 2022; Li et al., 2021), we utilized the Dirichlet distribution $p_k \sim Dir_N(\beta)$ to create non-IID data partitions across devices. For our experiments, we set $\beta = 0.1, 1, 5$. Clients were categorized into three types, namely small, medium, and large devices, following the approach outlined in (Liu et al., 2022). These devices were distinguished based on their computing power, with the ratios being $1 : 2 : 3$ respectively. It's worth noting that in our context, computing power strictly refers to the capacity of handling a model, while any discrepancies in computing speed were not taken into account.

**Baselines.** We consider two types of baselines: i) distillation-based methods (FedDF (Lin et al., 2020), DSFL(Itahara et al., 2021), FedET (Cho et al., 2022)) and ii) pruning-based methods (HeteroFL (Diao et al., 2021), Split-Mix (Hong et al., 2022), FedRolex (Alam et al., 2022)).

**Evaluation Metrics.** In our experiments, only the central server has the test dataset, and we use the global model to evaluate the performance of each method on the global test dataset. For fairness

---

**Algorithm 1:** Dynamic Federated Learning

---

**Input:** Initialized global model, $N$ edge devices with private datasets $\{\mathcal{D}_i\}_{i=1}^{N}$, communication round
number $T$, learning rate $\eta$, epoch number $E$.
**Output:** The final global model $\omega_g^T$.
**for** each communication round $t = 1, \ldots, T$ **do**
    Send the global model $\omega_g^{(t-1)}$ to the edge devices
    **for** each device $i = 1, 2, \ldots, N$ **in parallel do**
        Receive the global model $\omega_g^{(t-1)}$
        $\zeta_i^{(t)} \leftarrow$ **HeterogeneousLocalTraining** $(i, \omega_g^{(t-1)})$
        Re-parameterization to global model: $\omega_i^{(t)} \leftarrow \text{REP}(\zeta_i^{(t)}, \omega_g^t)$
        Return $\omega_i^{(t)}$ to server
    **end**
    $\omega_g^{(t)} \leftarrow \frac{1}{N} \sum \omega_i^{(t)}$
**end**
return $\omega_g^T$
**HeterogeneousLocalTraining** $(i, \omega_g^{(t-1)})$:
    Re-parameterization to local model: $\epsilon_i^{(t)} \leftarrow \text{REP}(\omega_g^{(t-1)}, \zeta_i^{(t-1)})$
    Train one local epoch: $\epsilon_i^{(t)} \leftarrow \epsilon_i^{(t)} - \eta \nabla \ell(\epsilon_i^{(t)}; \mathcal{D}_i)$
    Record gradient: $S_i^g \leftarrow \nabla \ell(\epsilon_i^{(t)}; \mathcal{D}_i)$
    Re-parameterization according to score $S_i^g, S_i^l$: $\zeta_i^{(t)} \leftarrow \text{DYMM}(\epsilon_i^t; S_i^g, S_i^l)$
    **for** local epoch $k = 1, 2, \ldots, E - 1$ **do**
        $\zeta_i^{(t)} \leftarrow \zeta_i^{(t)} - \eta \nabla \ell(\zeta_i^{(t+1)}; \mathcal{D}_i)$
        **if** $k = E - 1$ **then**
            Record gradient: $S_i^l \leftarrow \nabla \ell(\zeta_i^{(t)}; \mathcal{D}_i)$
        **end**
    **end**

---

evaluation, we used local heterogeneous models to compare accuracy on the global test set, which means that all accuracies in this paper are obtained on the global test set.

**Implementation Details.** We use PyTorch (Paszke et al., 2019) to implement DynamicFL and the other baselines. Unless otherwise specified, the number of local epochs is set to 3, we use the SGD optimizer with a local learning rate 0.01 for all approaches. The number of communication rounds is set to 200. The experiments involve 90 devices, and the sampling rate is set at 10%. We conducted our experiments on 4 NVIDIA RTX3090 GPUs.

### 3.1 PERFORMANCE EVALUATION

In our performance evaluation, we compare DynamicFL against state-of-the-art model-heterogeneous FL methods based on network pruning and knowledge distillation (KD). To ensure a fair comparison, we adopt the same experimental settings as previous studies, where the client model capacities are sampled from a uniform distribution.

**Static Scenario.** As shown in the Table 1, our method outperforms the state-of-the-art pruning-based baseline FedRolex by 6.19% and the distillation-based baseline FedET by 6.95% on CIFAR-10. To validate the generalization ability of our method on different network structures, we also evaluated on GoogleNet and LeNet. The results demonstrate that our method still achieves the highest accuracy on different network backbones. On LeNet-5, the accuracy of our method is 79.33%, 3.52% higher than the best performing baseline FedET. The superiority of our method is not limited to natural image datasets. On the medical image dataset DermaMNIST, the accuracy of our method is 71.48%, while the pruning-based FedRolex achieves an accuracy of only 66.76%. This confirms that our method can effectively modulate models of varying complexity to maximize performance for heterogeneous devices. It generalizes well to different neural network architectures and also medical imaging classification tasks, outperforming state-of-the-art alternatives in both natural and medical domains.

**Dynamic Scenario.** In practical federated learning scenarios, it is crucial to consider the dynamic nature of computing resources on participating devices. To comprehensively evaluate the performance

Table 1: Performance evaluation of test accuracy (%) with fixed computational budget.

| Model | | Method | CIFAR-10 | CIFAR-100 | DermaMNIST |
|---|---|---|---|---|---|
| LeNet-5 | KD-based | DSFL (Itahara et al., 2021) | 69.88 | 51.12 | 57.39 |
| | | FedDF (Lin et al., 2020) | 73.57 | 51.59 | 60.28 |
| | | FedET (Cho et al., 2022) | 75.81 | 52.43 | 63.73 |
| | Pruning-based | HeteroFL (Diao et al., 2021) | 71.39 | 50.67 | 58.77 |
| | | SplitMix (Hong et al., 2022) | 72.08 | 47.28 | 59.96 |
| | | FedRolex (Alam et al., 2022) | 74.72 | 52.81 | 60.13 |
| | Ours | DynamicFL | **79.33** | **63.08** | **66.44** |
| GoogLeNet | KD-based | DSFL (Itahara et al., 2021) | 70.35 | 52.61 | 62.97 |
| | | FedDF (Lin et al., 2020) | 74.34 | 53.95 | 63.82 |
| | | FedET (Cho et al., 2022) | 78.77 | 54.32 | 67.49 |
| | Pruning-based | HeteroFL (Diao et al., 2021) | 72.93 | 52.74 | 61.53 |
| | | SplitMix (Hong et al., 2022) | 73.14 | 50.69 | 62.58 |
| | | FedRolex (Alam et al., 2022) | 77.85 | 54.28 | 64.79 |
| | Ours | DynamicFL | **82.84** | **61.52** | **68.43** |
| ResNet-18 | KD-based | DSFL (Itahara et al., 2021) | 71.31 | 53.64 | 60.46 |
| | | FedDF (Lin et al., 2020) | 75.13 | 54. 37 | 62.73 |
| | | FedET (Cho et al., 2022) | 79.87 | 56.72 | 65.79 |
| | Pruning-based | HeteroFL (Diao et al., 2021) | 74.54 | 53.81 | 61.10 |
| | | SplitMix (Hong et al., 2022) | 75.21 | 52.27 | 62.48 |
| | | FedRolex (Alam et al., 2022) | 80.63 | 55.26 | 66.76 |
| | Ours | DynamicFL | **86.82** | **63.27** | **71.48** |

of various heterogeneous FL methods under these dynamic conditions, we examined four settings: static computing power, increasing computing power, decreasing computing power, and randomly changing computing power. Specifically, during the mid-phase of each local training cycle (i.e., after the completion of the second local epoch), we simulated a 5% change in the device's computing power. Knowledge distillation-based methods face challenges in adapting to these dynamic shifts in device computing power. They are constrained to re-request adapted models from the central server, resulting in significant delays in the current device's computing process. Pruning-based methods offer some adaptability by selectively pruning channels of the model to accommodate scenarios with decreasing computing resources. However, they struggle to cope with situations involving increasing computing resources. In contrast, our method exhibits remarkable flexibility in adapting to these dynamic changes in computing resources. For instance, as demonstrated in Table 2, when computing power decreases, our method achieves an accuracy that is 8.68% higher than that of FedRolex. Moreover, in scenarios where device computing power increases, our method outperforms other methods, achieving an accuracy of 88.62%—a 1.80% improvement over the fixed computing power setting. This underscores the robustness and adaptability of our approach in dynamic computing environments.

Table 2: Performance evaluation of test accuracy (%) with dynamic computational budget.

| | Method | Dynamic Resource | | | |
|---|---|---|---|---|---|
| | | Static | Decreasing | Increasing | Random |
| KD-based | DSFL (Itahara et al., 2021) | 71.31 | N/A | N/A | N/A |
| | FedDF (Lin et al., 2020) | 75.13 | N/A | N/A | N/A |
| | FedET (Cho et al., 2022) | 79.87 | N/A | N/A | N/A |
| Pruning-based | HeteroFL (Diao et al., 2021) | 74.54 | 70.07 | N/A | N/A |
| | SplitMix (Hong et al., 2022) | 75.21 | 70.71 | N/A | N/A |
| | FedRolex (Alam et al., 2022) | 80.63 | 75.68 | N/A | N/A |
| Ours | DynamicFL | **86.82** | **84.36** | **88.62** | **84.36** |

## 3.2 EVALUATION ON FAIRNESS

We assessed the level of fairness exhibited by the compared heterogeneous FL methods by examining their performance disparities on indigent, medium, and strong devices. As illustrated in Fig. 2, both distillation-based and pruning-based methods demonstrate markedly lower performance on indigent devices in comparison to strong devices. For example, HeteroFL achieves an accuracy of 71.31% on indigent devices, 76.8% on medium devices, and 82.87% on strong devices. In sharp contrast, our method attains a uniform accuracy of 88.24% across indigent, medium, and strong devices. This

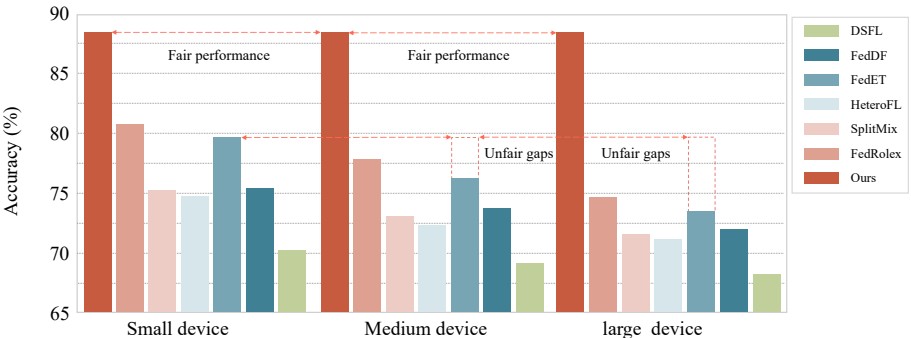

Figure 2: Fairness Evaluation. Our method can achieve the same accuracy in small and large device in the global test dataset.

uniformity arises from the assurance provided by re-parameterization, which ensures that the global model and the modulated local models yield identical outputs. This comparison substantiates that our approach not only achieves high accuracy but also upholds fairness in the process.

### 3.3 IMPACT OF THE DATASET DISTRIBUTION

We investigated the effect of data heterogeneity by manipulating the concentration parameter $\beta$ of the Dirichlet distribution on the CIFAR-10 and CIFAR-100 datasets. A smaller $\beta$ value resulted in a more imbalanced data partition. As shown in Table 3, DynamicFL consistently achieved the highest accuracy across the three imbalance levels. When $\beta = 0.5$, indicating the highest heterogeneity, our method attained an accuracy of 78.19% on CIFAR-100, surpassing the highest accuracy of 75.83% attained by other methods by a substantial margin of 2.64%. When data heterogeneity $\beta = 5$, our method still achieved the highest accuracy of 86.82%. The experiments demonstrated the effectiveness and robustness of the proposed DynamicFL framework. It consistently outperformed existing methods under different degrees of non-IID data.

Table 3: Influence of the Dirichlet distribution coefficient $\beta$.

|  | Method | $\beta$ | | |
| --- | --- | --- | --- | --- |
|  |  | 0.5 | 1 | 5 |
| | DSFL (Itahara et al., 2021) | 69.49 | 70.49 | 71.31 |
| KD-based | FedDF (Lin et al., 2020) | 70.17 | 73.14 | 75.13 |
| | FedET (Cho et al., 2022) | 75.83 | 78.89 | 79.87 |
| | HeteroFL (Diao et al., 2021) | 69.46 | 72.41 | 74.54 |
| Pruning-based | SplitMix (Hong et al., 2022) | 70.13 | 73.59 | 75.21 |
| | FedRolex (Alam et al., 2022) | 74.27 | 76.51 | 80.63 |
| Ours | DynamicFL | 78.19 | 83.44 | 86.82 |

## 4 CONCLUSION

In this paper, we presented DynamicFL, a novel framework designed for federated learning amidst heterogeneous computing resources across clients. Our approach employs structural re-parameterization to enable adaptive modulation of local models. During local training at each client, DynamicFL exhibits the capability to dynamically adjust the size of the local model based on the available computational resources, thus maximizing the utilization of the existing computing power. This addresses the limitations of current methods relying on knowledge distillation and network pruning, which face challenges related to computational efficiency and model mismatch. Furthermore, we provide a theoretical convergence analysis for our proposed method. To summarize, our work represents a significant stride towards realizing practical federated learning in real-world scenarios characterized by diverse computing resources. DynamicFL offers an elegant solution for resource-constrained federated learning, opening up opportunities for broader applications of this technology.

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

## A  CONVERGENCE ANALYSIS OF DYNAMICFL

We provide a detailed convergence analysis of DynamicFL in the supplementary material. Convergence analysis of federated learning has been extensively studied. Our method differs from standard federated learning in that during each local training phase, the convolutional modules of the global model are reparameterized into a multi-branch structure to adapt to the heterogeneous computing power of different devices. The main inspiration behind our proof is that since the multi-branch convolutions formed by reparameterization during inference can be re-fused into a single convolutional kernel, this equivalence should still hold during training. Therefore, the core idea of our proof is to posit a virtual sequence $\phi_i^{(t)}$ on the local device with the same structure as the global model and analyze the difference between $\omega_i^{(t)}$, obtained by reparameterizing the global model $\omega_g^{(t-1)}$ into the local model $\zeta_i^{(t)}$ on the local device and then reparameterizing it back, to obtain the convergence of the global model.

**Assumption 1** ($L$-smoothness and $\sigma$-uniformly bounded gradient variance).

*(a) F is L-smooth, i.e., $F(u) \leq F(x) + \langle \nabla F(x), u - x \rangle + \frac{1}{2}L \|u - x\|^2$ for any $u, x \in \mathbb{R}^d$.*

*(b) There exists a constant $G_{max} > 0$ such that: $\mathbb{E}\left[||\nabla F^{(i)}(x)||^2\right] \leq G_{max}^2, \quad \forall i \in [N], \forall \mathbf{x} \in \mathbb{R}^d$, where $\nabla F^{(i)}(x)$ is an unbiased stochastic gradient of $f^{(i)}$ at $x$.*

*(c) $\nabla f(x)$ has $\sigma^2$ -bounded variance, i.e., $\mathbb{E}_{\xi \sim S_i} \|\nabla F_i(\mathbf{x}) - \nabla f_i(\mathbf{x})\| \leq \sigma^2, \quad \forall i \in [N], \forall \mathbf{x} \in \mathbb{R}^d$.*
**Lemma 2** *For any reparameterization of a convolutional layer $l$ that can be represented as a summation of $N$ convolutional branches with weights $W_{l,n}^{(t)}$ and binary receptive field mask $M_{l,n}$, for $n = 1, \ldots, N$, its gradient descent update can be respresented as:*

$$W_l^{(t+1)} \Leftarrow W_l^{(t)} - \lambda^{(t)} f \left( \mathcal{G}_l \odot \frac{\partial \mathcal{L}}{\partial W_l^{(t)}}, W_l^{(t)}, \ldots, \mathcal{G}_l \odot \frac{\partial \mathcal{L}}{\partial W_l^{(0)}}, W_l^{(0)} \right), \tag{17}$$

where $\mathcal{G}_l = \sum_{n=1}^{N} M_{l,n}$.

**Proof:**

In DynamicFL, it can be expressed that for the local model $\zeta_{i,k}^{(t)}$ on edge device $i$,:

$$\zeta_{i,k+1}^{(t)} \leftarrow \zeta_{i,k}^{(t)} - \eta \nabla \ell(\zeta_i^{(t+1)}; \mathcal{D}_i). \tag{18}$$

We posit that after each batch training, the local model $\zeta_{i,k+1}^{(t)}$ is transformed to attain the same architecture as the global model $\omega_g$, , thereby obtaining $\omega_{i,k+1}^{(t)}$.

We also posit a virtual sequence $\phi_{i,k+1}^{(t)}$ which model architecture is the same as the global model. If it were to undergo local training on the local device, its corresponding gradient would be $\nabla \ell(\phi_i^{(t+1)}; \mathcal{D}_i)$. To render it the same as $\omega_{i,k+1}^{(t)}$, we would need to apply a spatially-specific rescaling to its gradient, with the scaling factor being $\mathcal{G}_i^{(t)}$, yielding:

$$\phi_{i,k+1}^{(t)} \leftarrow \phi_{i,k}^{(t)} - \mathcal{G}_i^{(t)} \odot \eta \nabla \ell(\phi_i^{(t+1)}; \mathcal{D}_i) = \omega_{i,k+1}^{(t)}. \tag{19}$$

Therefore, for each layer $W_l^{(t)}$ of $\phi_i^{(t)}$, we have:

$$
\begin{aligned}
W_l^{(t+1)} &\Leftarrow W_l^{(t)} - \lambda^{(t)} \mathcal{G}_l \odot \frac{\partial \mathcal{L}}{\partial W_l^{(t)}} \\
&\Leftarrow W_l^{(t)} - \lambda^{(t)} f \left( \mathcal{G}_l \odot \frac{\partial \mathcal{L}}{\partial W_l^{(t)}}, W_l^{(t)}, \ldots, \mathcal{G}_l \odot \frac{\partial \mathcal{L}}{\partial W_l^{(0)}}, W_l^{(0)} \right),
\end{aligned}
\tag{20}
$$

We arrive at lemma 2.

Therefore, reparameterization operations can be seen a spatial gradient scaling applied to the original convolution. Here we assume $\mathcal{G}_l \leq \mathcal{G}$. we can aggregate the virtual sequences $\phi_i^{(t)}$ on each device to obtain the global model $\omega_t^{(g)}$, which satisfies:

$$\mathbb{E}\left[\left\|\phi_{i,k}^{(t)} - \omega_t^{(g)}\right\|^2\right] \leq \mathbb{E}\left[\left\|\phi_{i,k}^{(t)} - \omega_{i,k}^{(t)} + \omega_{i,k}^{(t)} - \omega_t^{(g)}\right\|^2\right] \leq 4\eta^2 K^2 \mathcal{G}^2 G^2 \quad \forall t, k, i. \tag{21}$$

**Theorem 3** *The sequence generated by our method with stepsize $\eta \leq 1L$ satisfies*

$$\frac{1}{T}\sum_{t=1}^{T}\mathbb{E}\left[\left\|\nabla f\left(w_t^{(g)}\right)\right\|^2\right] \leq \frac{2}{\eta T}\left(\mathbb{E}\left[f\left(w_1^{(g)}\right)\right] - f\left(w_T^{(g)}\right)\right) + 4\eta^2 \mathcal{G}^2 L^2 G^2 K^2 + \frac{L}{N}\eta\sigma^2. \tag{22}$$

**Corollary 4.** When the function $f$ is lower bounded with $f\left(w_1^{(g)}\right) - f^* \leq \Delta$ and the number rounds $T$ is large enough, then set the stepsize $\eta = \frac{\sqrt{N}}{L\sqrt{T}}$ yields

$$\frac{1}{TK}\sum_{t=1}^{T}\sum_{k=1}^{K}\mathbb{E}\left[\left\|\nabla f\left(w_t^{(g)}\right)\right\|^2\right] = O\left(\frac{2L\Delta + \sigma^2}{\sqrt{NT}} + \frac{N}{T}\right). \tag{23}$$

The first term dominates so our algorithm shares the same convergence speed $O(1/\sqrt{NT})$ as the vanilla FedAvg.

**Proof:**

By the *L*-smoothness of $f$, we have:

$$\mathbb{E}\left[f\left(w_{t,k+1}^{(g)}\right)\right] \leq \mathbb{E}\left[f\left(w_{t,k}^{(g)}\right) + \left\langle\nabla f\left(w_{t,k+1}^{(g)}\right), w_{t,k+1}^{(g)} - w_{t,k}^{(g)}\right\rangle + \frac{L}{2}\left\|w_{t,k+1}^{(g)} - w_{t,k}^{(g)}\right\|^2\right]. \tag{24}$$

Note that

$$\mathbb{E}\left[\left\|w_{t,k+1}^{(g)} - w_{t,k}^{(g)}\right\|^2\right] = \eta^2\mathbb{E}\left[\left\|\frac{1}{N}\sum_{i=1}^{N}\mathcal{G}_{i,k}^{(t)}G_{i,k}^{(t)}\right\|^2\right]$$

$$= \eta^2\mathbb{E}\left[\left\|\frac{1}{N}\sum_{i=1}^{N}\left(\mathcal{G}_{i,k}^{(t)}G_{i,k}^{(t)} - \nabla f_i\left(w_{i,k}^{(t)}\right)\right)\right\|^2\right] + \eta^2\mathbb{E}\left[\left\|\frac{1}{N}\sum_{i=1}^{N}\nabla f_i\left(w_{i,k}^{(t)}\right)\right\|^2\right]$$

$$= \frac{\eta^2}{N^2}\sum_{i=1}^{N}\mathbb{E}\left[\left\|\mathcal{G}_{i,k}^{(t)}G_{i,k}^{(t)} - \nabla f_i\left(w_{l,k}^i\right)\right\|^2\right] + \eta^2\mathbb{E}\left[\left\|\frac{1}{N}\sum_{i=1}^{N}\nabla f_i\left(w_{l,k}^i\right)\right\|^2\right]$$

$$\leq \frac{\eta^2\sigma^2}{N} + \eta^2\mathbb{E}\left[\left\|\frac{1}{N}\sum_{i=1}^{N}\nabla f_i\left(w_{i,k}^{(t)}\right)\right\|^2\right]. \tag{25}$$

Moreover,

$$\mathbb{E}\left[\left\langle\nabla f\left(w_{t,k}^{(g)}\right), w_{t,k+1}^{(g)} - w_{t,k}^{(g)}\right\rangle\right]$$

$$= -\eta\mathbb{E}\left[\left\langle\nabla f\left(w_{t,k}^{(g)}\right), \frac{1}{N}\sum_{i=1}^{N}\mathcal{G}_{i,k}^{(t)}G_{i,k}^{(t)}\right\rangle\right]$$

$$= -\eta\mathbb{E}\left[\left\langle\nabla f\left(w_{t,k}^{(g)}\right), \frac{1}{N}\sum_{i=1}^{N}\nabla f_i\left(\phi_{i,k}^{(t)}\right)\right\rangle\right]$$

$$= \frac{\eta}{2}\mathbb{E}\left[\left\|\nabla f\left(w_{t,k}^{(g)}\right) - \frac{1}{N}\sum_{i=1}^{N}\nabla f_i\left(\phi_{i,k}^{(t)}\right)\right\|^2\right] - \frac{\eta}{2}\mathbb{E}\left[\left\|\nabla f\left(w_{t,k}^{(g)}\right)\right\|^2\right] - \frac{\eta}{2}\mathbb{E}\left[\left\|\frac{1}{N}\sum_{i=1}^{N}\nabla f_i\left(\phi_{i,k}^{(t)}\right)\right\|^2\right] \tag{26}$$

The first term can be upper bounded with the $L$-smoothness condition:

$$
\mathbb{E}\left[\left\|\nabla f\left(w_{t,k}^{(g)}\right) - \frac{1}{N}\sum_{i=1}^{N}\nabla f_i\left(\phi_{i,k}^{(t)}\right)\right\|^2\right] = \mathbb{E}\left[\|\frac{1}{N}\sum_{i=1}^{N}\left(\nabla f_i\left(w_{t,k}^{(g)}\right) - \nabla f_i\left(\phi_{i,k}^{(t)}\right)\right)\|^2\right]
$$

$$
\text{(Jensen's inequality )} \le \frac{1}{N}\sum_{i=1}^{N}\mathbb{E}\left[\|\left(\nabla f_i\left(w_{t,k}^{(g)}\right) - \nabla f_i\left(\phi_{i,k}^{(t)}\right)\right)\|^2\right] \quad (27)
$$

$$
\text{($L$-smoothness )} \le \frac{L^2}{N}\sum_{i=1}^{N}\left\|w_{t,k}^{(g)} - \phi_{i,k}^{(t)}\right\|^2
$$

$$
\le 4\eta^2\mathcal{G}^2 L^2 G^2 K^2
$$

Now, combining the above inequalities together with $\eta \le 1L$ yields

$$
\mathbb{E}\left[f\left(w_{t,k+1}^{(g)}\right)\right] - \mathbb{E}\left[f\left(w_{t,k}^{(g)}\right)\right]
$$

$$
\le \frac{L\eta^2 - \eta}{2}\mathbb{E}\left[\left\|\frac{1}{N}\sum_{i=1}^{N}\nabla f_i\left(\phi_{i,k}^{(t)}\right)\right\|^2\right] - \frac{\eta}{2}\mathbb{E}\left[\left\|\nabla f\left(w_{t,k}^{(g)}\right)\right\|^2\right] + 2\eta^3 L^2 G^2(K+D)^2 + \frac{L}{2N}\eta^2\sigma^2
$$

$$
\le -\frac{\eta}{2}\mathbb{E}\left[\left\|\nabla f\left(w_{t,k}^{(g)}\right)\right\|^2\right] + 2\eta^3 L^2 \mathcal{G}^2 G^2 K^2 + \frac{L}{2N}\eta^2\sigma^2.
$$

$$(28)$$

Rearrange the terms yields:

$$
\mathbb{E}\left[\left\|\nabla f\left(w_{t,k}^{(g)}\right)\right\|^2\right] \le \frac{2}{\eta}\left(\mathbb{E}\left[f\left(w_{t,k}^{(g)}\right)\right] - \mathbb{E}\left[f\left(\overline{w_{t,k+1}}\right)\right]\right) + 4\eta^2 L^2 \mathcal{G}^2 G^2 K^2 + \frac{L}{N}\eta\sigma^2. \quad (29)
$$

Finally,

$$
\frac{1}{TK}\sum_{t=1}^{T}\sum_{k=1}^{K}\mathbb{E}\left[\left\|\nabla f\left(w_{t,k}^{(g)}\right)\right\|^2\right] \le \frac{2}{\eta T}\left(\mathbb{E}\left[f\left(w_1^{(g)}\right)\right] - f^*\right) + 4\eta^2 \mathcal{G}^2 L^2 G^2 K^2 + \frac{L}{N}\eta\sigma^2. \quad (30)
$$

