# OpenReview forum: "Everyone Counts: Fair and Accurate Heterogeneous Federated Learning with Resource-Adaptive Model Modulation"
_ICLR.cc/2024/Conference — ICLR 2024 Conference Withdrawn Submission_

### Official Review · Reviewer_UcXY · 2023-10-31

**Soundness:** 3 good
**Presentation:** 2 fair
**Contribution:** 2 fair
**Rating:** 3
**Confidence:** 4

**Summary:**

This work considers an approach in which resource-constrained devices participating in FL training can achieve seamless knowledge transfer between diverse heterogeneous models. The proposed Dynamic FL method does structural re-parameterization of the local model per device compute abilities. Experimental results show performance improvement as compared with standard baselines, such as HeteroFL and KD-based methods.

**Strengths:**

This work employs structural re-parameterization to enable adaptive modulation of local models, which is better for a practical FL setting where the collaborating nodes have heterogeneous computing resources. It has been shown this approach enables fair participation of clients with improved model accuracy (as compared to known methods to address heterogeneous FL, like KD and pruning). The authors have theoretically provided a convergence guarantee for their method.

**Weaknesses:**

I have concerns regarding the technical aspects of this work, as follows:

1. Several aspects are not clearly defined and discussed in the paper. For instance, after (2), what is r_i, and how is it defined?
2. As the structural re-parameterization is done each round (in my understanding), it certainly adds computational burden during training on the already constrained nodes. Then, the question arises: what is the cost of executing (4)?
3. The manuscript is not clear in justifying how, through (5), to assess the contribution of branches to the global aggregation. I feel the authors missed (in 2.2.2) building a clear connection between re-parameterization and local training operation during model training. For instance, how "contribution" is quantified per se, and is this only with the adaptation in the number of layers, and also the number of parameters per layer, and so on.
4. Is the number of epoch E fixed for each client? If so, why? because one may argue clients can be adaptive in choosing the number of epochs during local training.
5. It is unclear why the aggregation scheme, as in (11), was used. Also, can the authors please justify further their claim: "operations in re-parameterization guarantee lossless knowledge transfer across heterogeneous local models"?
6. While the experiments cover a broad range of (relevant) baselines (which is commendable), however, I am not sure the comparison is sufficient to demonstrate improvement. For instance, the authors have not mentioned why the number of local epochs was set (fixed) to 3 (for all clients); the model capacity is only abstracted without particular practical details - how do make these changes in the computing abilities; no convergence analysis (experimental) is provided, which can be added in my understanding; training time-complexity analysis is missing (also, what is the training efficiency), and so on. In fairness analysis, how is the accuracy reported?

Minor:
- Figure 1 is unclear in its current form, and the description is insufficient. How to interpret this? The representative Diverse Branch Block (DBB), as in [Ding et al., 2021a,b], is unclear to the readers unfamiliar for readers.
- notations are a bit difficult to follow.

**Questions:**

Please kindly see the comments in the weakness section.

---

### Official Review · Reviewer_Mgiw · 2023-10-31

**Soundness:** 3 good
**Presentation:** 3 good
**Contribution:** 3 good
**Rating:** 5
**Confidence:** 3

**Summary:**

The authors introduce what they call DynamicFL: the idea is to make federated learning practical for heterogeneous set of devices by dynamically adjusting the model architecture (re-parameterization) during local training. Each client will then train a slightly different architecture of the model, depending on the available resources. Afterwards, the model will be scaled back and shared for aggregation.

**Strengths:**

- Heterogeneous FL is important, especially when we need to train larger models on a wide range of end-clients
- The paper is well presented and written
- The idea of structural re-parameterization is interesting contribution

**Weaknesses:**

The paper is interesting to read, but there are some areas to improve:

- The fact that the model is always scaled up (on the clients) and then scaled down before aggregating is a bit counter intuitive. One would expect that the aggregated (server-side) model would need additional capacity to learn from the wide distribution of data, rather than the device submodes. This might restrict the overall capacity of the final aggregated model. Maybe the authors can further motivate this approach.

- Similarly, this paper focuses on convolutions, limiting a bit the applicability on other architectures

- The authors tackle a heterogeneous environment with device of different capabilities. In most such use-cases devices typically are sampled and are likely to participate for just a few rounds (in some use-cases just once) during the FL training. I was wondering what would be the impact of having clients participating only for 1 or K (where K is small) rounds.

- The evaluation does not provide any insights on energy, memory footprint, transfer data volume, convergence speed. This is important as this method is focusing towards performance on a heterogeneous set of devices.

**Questions:**

- The fact that we scale up the model on-device, does this mean that the largest possible model we can support for inference is determined by the weakest devices ?

- In the experiments, the authors show that other methods (HeteroFL , Split Mix etc) have worse accuracy. But a benefit of these approaches is that they can scale *down* the model (I.e., submode training). Whereas dynamicFL practically scales up. I was wondering if the authors used a larger initial (aggregated) architecture for these models when compared to their method. Essentially, a fair comparison would be having the other methods provision for the large devices at the aggregator level (I.e., having a larger initial model).

---

### Official Review · Reviewer_nk4n · 2023-11-01

**Soundness:** 2 fair
**Presentation:** 2 fair
**Contribution:** 2 fair
**Rating:** 5
**Confidence:** 3

**Summary:**

This paper presents Dynamic Federated Learning (DynamicFL), addressing the challenge of heterogeneous edge devices in federated learning. DynamicFL employs a structural re-parameterization to adapt local models, ensuring a fair client participation while achieving a high test accuracy. It outperforms existing methods, including knowledge distillation and network pruning, in heterogeneous FL scenarios.

**Strengths:**

1. The paper is easy to follow.
2. The authors conducted a series of comparative experiments involving several distinct methods.

**Weaknesses:**

1. The major issue of this paper is about the technical contribution. The techniques employed in this paper, specifically the 'REF' and 'DYMM,' appear to be exsiting ones proposed in prior works, which could potentially diminish the originality and contribution of this article.
2. The proposed theoretical framework focuses primarily on the analysis of convergence rates, yet the experimental part lacks the presentation of any convergence curves.

**Questions:**

Can the authors provide an analysis that quantifies the additional computational and storage resources consumed by the introduced operations 'REF' and 'DYMM'?

---

### Public Comment · ~Saurav_Prakash1 · 2023-11-21
**Closely related work**

Dear Authors,


I found your work to be interesting. In this regard, I would also like to draw your attention to our following related work, where we proposed PriSM methodology to extract small sub-models, while maintaining a near full coverage of the FL server model. As we show in the paper, we perform significantly better in comparison to existing SOTA, particularly in the regime when each client is extremely resource constrained. It would be good to see comparisons of your approach with PriSM.


* Niu, Y., Prakash, S., Kundu, S., Lee, S. and Avestimehr, S., 2023. “Overcoming Resource Constraints in Federated Learning: Large Models Can Be Trained with only Weak Clients”. Published in Transactions on Machine Learning Research, 2023 (link: https://openreview.net/forum?id=lx1WnkL9fk). Partly presented at the NeurIPS Workshop on Recent Advances and New Challenges in Federated Learning (FL-NeurIPS), 2022 (link: https://openreview.net/forum?id=e97uuEXkSii).


Thanks and Regards,


Saurav